

# Decrease in the usual walking speed and body fat percentage associated with a deterioration in long-term care insurance certification levels

Yohei Sawaya[1,2,*], Tamaki Hirose[1,2,*], Takahiro Shiba[3], Ryo Sato[2], Lu Yin[3], Akira Kubo[4] and Tomohiko Urano[2,3,5]

[1] Department of Physical Therapy, School of Health Sciences, International University of Health and Welfare, Otawara, Tochigi, Japan
[2] Nishinasuno General Home Care Center, Department of Day Rehabilitation, Care Facility for the Elderly "Maronie-en", Nasushiobara, Tochigi, Japan
[3] Integrated Facility for Medical and Long-term care, Care Facility for the Elderly "Maronie-en", Nasushiobara, Tochigi, Japan
[4] Department of Physical Therapy, School of Health Sciences at Odawara, International University of Health and Welfare, Odawara, Kanagawa, Japan
[5] Department of Geriatric Medicine, School of Medicine, International University of Health and Welfare, Narita, Chiba, Japan
* These authors contributed equally to this work.

Corresponding authors
Yohei Sawaya, sawaya@iuhw.ac.jp, yohei.sawaya@gmail.com
Tomohiko Urano, turano@iuhw.ac.jp

## ABSTRACT

**Background**. In Japan, the number of older adults requiring long-term care insurance (LTCI) is increasing and the cost is becoming a social problem. In these fields, the role of geriatric rehabilitation includes maintaining the physical function and LTCI certification levels. The prevalence of sarcopenia is high among older adults requiring LTCI certification, and there are many opportunities to assess the handgrip strength, walking speed, and muscle mass. This study aimed to identify sarcopenia-related assessments sensitive to transitions in LTCI certification levels and determine cut-off values to predict them.

**Methods**. This prospective cohort study analyzed 98 daycare users (mean age ± standard error: 78.5 ± 0.8 years) between March 2019 and 2023. The participants received LTCI certification before the study, and their levels were renewed between baseline and follow-up (six months later). The measurements included handgrip strength, usual walking speed, body composition, and SARC-F score. Participants were classified into maintenance, deterioration, and improvement groups according to the changes in their LTCI certification levels. We identified factors contributing to the deterioration of LTCI certification levels using baseline and before and after comparisons, multivariate analyses, and receiver operating characteristic analyses.

**Results**. No significant differences were observed in the baseline data among the groups. Only the deterioration group showed significant changes in the usual walking speed (baseline: 0.64 ± 0.25 m/s, follow-up: 0.53 ± 0.21 m/s, $P = 0.008$) and body fat percentage (baseline: 29.2 ± 9.9%, follow-up: 27.7 ± 10.3%, $P = 0.047$). Binomial logistic regression showed that changes in usual walking speed ($P = 0.042$) and body fat percentage ($P = 0.011$) were significantly associated with the deterioration of LTCI certification levels, even after adjustment. The cutoff values of change to discriminate

the deterioration of LTCI certification levels were −0.14 m/s at the usual walking speed ($P = 0.047$) and −1.0% for body fat percentage ($P = 0.029$).

**Conclusions**. Decreases in usual walking speed and body fat percentage may predict worse certification levels in older adults requiring LTCI.

# INTRODUCTION

The long-term care insurance (LTCI) system was established in 2000 to support the care of older adults in Japan, which has the largest older population globally. The number of individuals receiving LTCI certification increased from 4,907,000 in 2010 to 6,689,000 in 2020 (*Cabinet Office, 2023*). The cost of LTCI increased from 3.6 trillion Japanese Yen (US$ 32.7 billion) when the system was introduced in 2000, to 11.7 trillion Japanese Yen (US$ 106.4 billion) in 2019 (*Yamada & Arai, 2020*). Under the LTCI system, eligible individuals receive long-term care benefits according to seven levels of certification by estimated daily care minutes (*Ministry of Health, Labor, and Welfare, 2021*). Higher estimated daily care minutes are marked with more severe certification levels, and the maximum benefits covered by the LTCI are higher for severe certification levels. These certification levels are periodically updated. The main services include visiting, commuting, short stay, and nursing homes, and their availability and frequency differ depending on the certification level (*Konishi, Inokuchi & Yasunaga, 2024*). Therefore, changes in LTCI certification levels reflect the physical and mental functions as well as health status, with significant social and economic implications. As the population ages and the number of people needing long-term care increases, the sustainability of insurance finances becomes a risk. Maintaining and improving the health of those eligible for LTCI to reduce long-term care costs has become an important policy issue (*Yamada & Arai, 2020*; *Konishi, Inokuchi & Yasunaga, 2024*).

Previous studies on the factors contributing to LTCI have mainly followed older adults with no applicable LTCI for several years, searching for factors that contribute to the development of new LTCI needs (*Shimada et al., 2023*; *Ashida, Kondo & Kondo, 2016*; *Shimizu, Ide & Kondo, 2023*; *Matsumoto & Takatori, 2021*). Previous studies analyzing changes in certification levels among LTCI recipients have several limitations regarding the assessment items and target individuals, including analyses focusing solely on basic characteristics (*Kumai & Ikeda, 2021*; *Ohnuma et al., 2016*), analyses based on several basic characteristics and one functional assessment (*Kamiya et al., 2017*), analyses combined participants with no applicable LTCI and those who already had LTCI certification (*Kawamura et al., 2021*), and analyses of institutionalized residents (*Jin et al., 2018*). Thus, there exists a lack of studies addressing the factors contributing to changes in LTCI certification-level renewal in terms of sarcopenia assessment, targeting older adults who originally received LTCI certification and required assistance or care while receiving care and living at home. Sarcopenia is a functional decline associated with the loss of

skeletal muscle mass and has been a major focus in geriatrics in recent years owing to its strong association with decreased quality of life and increased incidence of falls and fractures (*Chen et al., 2020*; *Beaudart et al., 2015*; *Yu, Leung & Woo, 2014*). In Japan, the prevalence of sarcopenia is 9.9% among older adults with no applicable LTCI, while it ranges 41.7–60.2% among those requiring LTCI, which is extremely high. (*Makizako et al., 2019*; *Kitamura et al., 2021*; *Kamo et al., 2018*; *Sawaya et al., 2020*). Furthermore, several reports have indicated that older adults with sarcopenia requiring LTCI tend to have severe certification levels, fewer daily steps, and an association between the certification levels and the SARC-F score (*Yin et al., 2023*; *Kitamura et al., 2021*; *Tsugihashi et al., 2021*). Considering this background, frequent assessment of sarcopenia among older adults requiring LTCI is essential in geriatric rehabilitation. Therefore, determining a simple screening method to efficiently predict and detect changes in LTCI certification levels based on sarcopenia assessment and recommending effective rehabilitation interventions to maintain or improve LTCI certification levels is beneficial.

In this study, we hypothesized that sarcopenia-related assessments, which are often associated with negative outcomes, are also associated with changes in LTCI certification levels. Our study aimed to address the following research questions: (1) we explored which sarcopenia-related assessments demonstrate sensitivity to changes in the LTCI certification levels; (2) we investigated which among these assessments, baseline data, or degree of change reflects a shift in the LTCI certification levels; and (3) we sought to establish a cut-off value for these assessments in understanding and predicting changes in LTCI certification levels.

## MATERIALS & METHODS

### Study design
This single-center prospective cohort study was conducted between March 2019 and March 2023. Participants were recruited through verbal announcements and posts at the facility. All participants were verbally informed about the study and provided written informed consent. This study was approved by the Ethical Review Committee of the International University of Health and Welfare (approval numbers 17-Io-189-7 and 21-Io-22-2) and adhered to the tenets of the Declaration of Helsinki.

### City information
This study was conducted in Nasushiobara City, Tochigi Prefecture, Japan (https://www.city.nasushiobara.tochigi.jp/index.html). Nasushiobara City is located in the northern part of the Tochigi Prefecture, approximately 150 km from Tokyo. As of April 1, 2023, the population was 114,334, with a density of 192.89 people/km$^2$. It was classified as suburban according to a previous study (*Hirose et al., 2024*). The participants lived in Nasushiobara City or nearby cities and towns.

### Study participants
The participants in this study were users of day care for older adults based on the Japanese LTCI system. Daycare is a rehabilitation service involving transportation. The rehabilitation

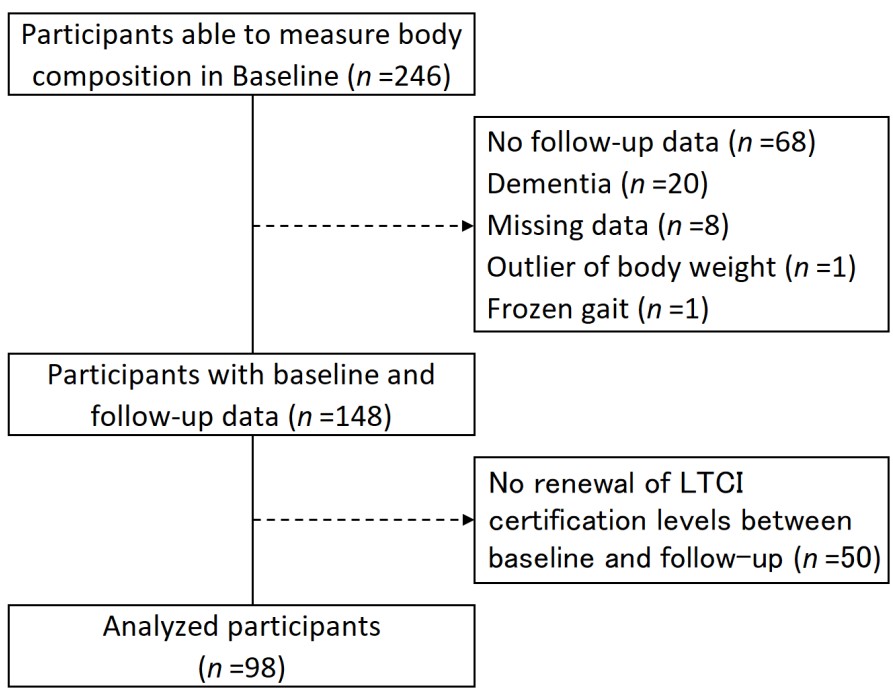

**Figure 1   Flowchart for participant intake.** LTCI, long-term care insurance.

program comprised muscle-strengthening exercises (using machine training), exercise therapy (employing a stationary bicycle ergometer), guidance on activities of daily living, exercise instructions, and others tailored to an individual's physical and mental conditions. All participants were assessed for LTCI certification level and sarcopenia at baseline and follow-up (six months later). In accordance with previous studies, the inclusion criteria for this study were participants who used our day care and were able to undergo body composition measurements in a standing position (*Sawaya et al., 2022*). The exclusion criteria were as follows: no follow-up data, diagnosis of dementia, missing measurements, outliers (weight >100 kg), severely frozen gait, and no renewal of the LTCI certification level between baseline and follow-up. A flowchart of the participants is shown in Fig. 1. Physiotherapists performed all assessments.

**Long-term care insurance system**

The LTCI was administered through a national program in Japan. To be eligible for LTCI services, individuals must be certified for long-term care or support. Certification levels were determined based on the time required for an individual to be cared for by a computer system during the initial assessment, and by trained local government officials during the final assessment (*Yamada & Arai, 2020*). A written statement from the primary care doctor was also required for this application. The certification levels were divided into seven categories (support levels 1 and 2 and care-need levels 1–5), depending on the time required to care for these individuals (*Yamada & Arai, 2020*). The ordinal scales in this study were 1, 2, 3, 4, 5, 6, and 7 for support level 1 (least disabled), support level 2,

care-need level 1, care-need level 2, care-need level 3, care-need level 4, and care-need level 5 (most disabled).

## Changes in the long-term care insurance certification levels

The primary outcome of the study was a change in the LTCI certification levels. All participants were certified for LTCI before the study began and their LTCI certification levels were updated between baseline and follow-up. The deterioration and improvement groups were defined as those who experienced a change in one or more certification levels. The maintenance group included those who showed no change in their LTCI certification levels after renewal.

## Body composition

Body composition was measured in the standing position using a multifrequency bioelectrical impedance analysis (BIA) body composition analyzer (InBody 520, InBody Japan Inc., Japan). Body weight, body mass index, body fat mass, body fat percentage, the amount of skeletal muscle in the limbs divided by the square of height (skeletal muscle mass index; SMI), protein, and mineral were used as indices.

## Handgrip strength

Handgrip strength was measured twice on each side in a seated position using a Smedley grip strength meter (TKK 5401 Grip-D; Takei Scientific Instruments, Niigata, Japan) according to the Clinical Guidelines for Sarcopenia. The maximum value of the measurements was considered the representative value (*Arai, 2018*).

## Usual walking speed

The usual walking speed was measured once in a 5-m walk test with acceleration and deceleration paths before and after the measurement section according to the Clinical Guidelines for Sarcopenia (Arai, 2018). A stopwatch was used for all measurements, and the verbal instruction, ''walk at your normal speed as usual'', was provided before the measurement. Participants who used mobility aids used their usual mobility aids.

## SARC-F

The SARC-F is a five-question screening tool for sarcopenia (*Mo et al., 2021*). The physiotherapists individually asked each participant questions.

## Diagnosis of sarcopenia

Sarcopenia was diagnosed if the participants had low skeletal muscle mass, muscle strength, and/or physical function, according to the Asian Working Group for Sarcopenia 2019 algorithm (*Chen et al., 2020*).

## Statistical analyses

The baseline information was compared in the maintenance, deterioration, and improvement groups using one-way ANOVA, Kruskal-Wallis, $\chi^2$, and Fisher's exact tests. The baseline and follow-up measurements were subsequently compared using a paired $t$-test and Wilcoxon signed-rank test for the maintenance, deterioration, and improvement

groups. A binomial logistic regression analysis was performed with the deterioration of LTCI certification levels as the dependent variable (deterioration = 1; maintenance/improvement = 0) and changes in usual walking speed or body fat percentage, which showed significant changes between baseline and follow-up in the deterioration group, were the independent variables. The analysis was performed using two models: non-adjusted and adjusted for sex, LTCI certification level, and intractable neurological diseases. Power analyses were performed on both models using the post-hoc method. Given the sample size of this study, the only adjustment variables other than sex were LTCI certification level and intractable neurological diseases, which were significant in the baseline comparisons. Finally, the receiver operating characteristic (ROC) curve was analyzed using the Youden index method to calculate the area under the curve (AUC), sensitivity, specificity, and cutoff values for the change in usual walking speed and body fat percentage with deterioration in the LTCI certification levels (*Perkins & Schisterman, 2006*). Although this study was divided into three groups, binomial logistic regression and ROC were analyzed separately for two groups (deterioration, and maintenance group/improvement groups) (*Hayasaka et al., 2018*). Statistical analyses were performed using SPSS version 25 (IBM Japan, Tokyo, Japan) and G*Power version 3.1.9.2. (*Faul et al., 2007*). The significance level for statistical analysis was set at 5%.

# RESULTS

This study involved 98 participants, with an age (mean ± standard error) of 78.5 ± 0.8 years, consisting of 51 males and 47 females. Of the 98 participants analyzed in this study, 65 (66.3%), 21 (21.4%), and 12 (12.2%) belonged to the maintenance, deterioration, and improvement groups, respectively. Changes in the LTCI certification levels are shown in the raw data. The breakdown of the deterioration group was as follows: eight individuals experienced a one-stage deterioration, ten experienced a two-stage deterioration, two experienced a three-stage deterioration, and one experienced a four-stage deterioration. The breakdown of the improvement group was as follows: nine individuals experienced a one-stage improvement, two experienced a two-stage improvement, and one experienced a three-stage improvement. Table 1 compares the three groups at baseline. No significant differences among the three groups were observed in sarcopenia-related assessments, age, or body size at baseline; only LTCI certification level ($P = 0.004$) and intact neurological disease status ($P = 0.008$) were significantly different. Details of the chronic diseases of the participants is presented in Table S1. The deterioration group showed a significant decrease in the usual walking speed (baseline: 0.64 ± 0.25 m/s, follow-up: 0.53 ± 0.21 m/s, $P = 0.008$) and body fat percentage (baseline: 29.2 ± 9.9%, follow-up: 27.7 ± 10.3%, $P = 0.047$) (Table 2). No significant differences were observed between the maintenance and improvement groups.

Tables 3 and 4 show the factors associated with the deterioration of LTCI certification levels based on binomial logistic regression analysis. Changes in the usual walking speed were significantly associated with the deterioration of LTCI certification levels in the non-adjusted ($\beta = -4.558$, $P = 0.016$) and adjusted models ($\beta = -3.964$, $P = 0.042$). Similarly,

**Table 1 Baseline information for the maintenance, deterioration, and improvement groups.**

| | Maintenance Group (n = 65) | Deterioration group (n = 21) | Improvement group (n = 12) | P-value |
|---|---|---|---|---|
| Age (years) | 78.4 ± 7.7 | 80.1 ± 7.1 | 75.9 ± 11.2 | 0.362 |
| Sex, female, % | 52.3 | 42.9 | 33.3 | 0.419 |
| LTCI certification level (level) | 3 [1–3] | 2 [1–3] | 4 [3–4] | 0.004 |
| Height (cm) | 158.3 ± 8.4 | 157.2 ± 8.4 | 160.0 ± 8.8 | 0.664 |
| Body weight (kg) | 59.0 ± 9.7 | 55.6 ± 9.8 | 55.0 ± 10.4 | 0.223 |
| BMI (kg/m$^2$) | 23.5 ± 3.1 | 22.6 ± 3.9 | 21.5 ± 3.4 | 0.117 |
| Body fat mass (kg) | 18.1 ± 7.1 | 16.7 ± 7.7 | 15.5 ± 6.6 | 0.443 |
| Body fat percentage (%) | 30.0 ± 9.4 | 29.2 ± 9.9 | 27.6 ± 8.5 | 0.713 |
| Protein (kg) | 7.8 ± 1.3 | 7.4 ± 1.2 | 7.5 ± 1.4 | 0.465 |
| Mineral (kg) | 2.82 ± 0.45 | 2.66 ± 0.38 | 2.75 ± 0.48 | 0.339 |
| Handgrip strength (kg) | 22.7 ± 8.4 | 20.5 ± 9.5 | 21.5 ± 7.8 | 0.591 |
| Usual walking speed (m/s) | 0.76 ± 0.30 | 0.64 ± 0.25 | 0.66 ± 0.33 | 0.194 |
| SMI (kg/m$^2$) | 6.43 ± 0.94 | 6.16 ± 0.93 | 6.08 ± 1.05 | 0.331 |
| SARC-F (point) | 3 [1.5–5] | 5 [2–8] | 4 [3–5] | 0.100 |
| Sarcopenia, % | 44.6 | 57.1 | 75.0 | 0.126 |
| Cerebrovascular dis, % | 49.2 | 47.6 | 50.0 | 0.989 |
| Orthopedic dis, % | 58.5 | 52.4 | 41.7 | 0.539 |
| Cancer, % | 15.4 | 19.0 | 8.3 | 0.836 |
| Intractable neurological dis, % | 6.2 | 28.6 | 25.0 | 0.008 |

Notes.

The numbers in the table are presented in the following: mean ± standard deviation, %, and median [25th percentile–75th percentile].

LTCI certification levels were classified into seven levels (support levels 1 and 2 and care-need levels 1–5).

BMI, body mass index; dis, disease; LTCI, long-term care insurance; SMI, skeletal muscle mass index.

**Table 2 Changes in the baseline and follow-up measurements.**

| | Maintenance group (n = 65) | | | Deterioration group (n = 21) | | | Improvement group (n = 12) | | |
|---|---|---|---|---|---|---|---|---|---|
| | Baseline | Follow-up | P-value | Baseline | Follow-up | P-value | Baseline | Follow-up | P-value |
| Body weight (kg) | 59.0 ± 9.7 | 59.1 ± 9.4 | 0.690 | 55.6 ± 9.8 | 55.1 ± 10.9 | 0.253 | 55.0 ± 10.4 | 54.5 ± 10.6 | 0.425 |
| BMI (kg/m$^2$) | 23.5 ± 3.1 | 23.5 ± 3.0 | 0.703 | 22.6 ± 3.9 | 22.3 ± 4.4 | 0.161 | 21.5 ± 3.4 | 21.2 ± 3.4 | 0.339 |
| Body fat mass (kg) | 18.1 ± 7.1 | 18.4 ± 6.7 | 0.307 | 16.7 ± 7.7 | 15.9 ± 8.2 | 0.090 | 15.5 ± 6.6 | 15.1 ± 7.0 | 0.530 |
| Body fat percentage (%) | 30.0 ± 9.4 | 30.5 ± 9.0 | 0.164 | 29.2 ± 9.9 | 27.7 ± 10.3 | 0.047 | 27.6 ± 8.5 | 26.9 ± 9.2 | 0.527 |
| Protein (kg) | 7.8 ± 1.3 | 7.8 ± 1.3 | 0.524 | 7.4 ± 1.2 | 7.5 ± 1.3 | 0.652 | 7.5 ± 1.4 | 7.5 ± 1.3 | 0.921 |
| Mineral (kg) | 2.82 ± 0.45 | 2.83 ± 0.43 | 0.737 | 2.66 ± 0.38 | 2.68 ± 0.41 | 0.580 | 2.75 ± 0.48 | 2.71 ± 0.43 | 0.432 |
| Handgrip strength (kg) | 23.0 ± 7.9 | 23.2 ± 8.2 | 0.613 | 20.5 ± 9.5 | 19.7 ± 8.7 | 0.081 | 21.5 ± 7.8 | 22.7 ± 7.3 | 0.378 |
| Usual walking speed (m/s) | 0.76 ± 0.30 | 0.74 ± 0.28 | 0.207 | 0.64 ± 0.25 | 0.53 ± 0.21 | 0.008 | 0.66 ± 0.33 | 0.66 ± 0.31 | 0.904 |
| SMI (kg/m$^2$) | 6.43 ± 0.94 | 6.39 ± 0.93 | 0.397 | 6.16 ± 0.93 | 6.14 ± 0.88 | 0.845 | 6.08 ± 1.05 | 6.12 ± 1.07 | 0.578 |
| SARC-F (point) | 3 [1.5–5] | 3 [1–5] | 0.204 | 5 [2–8] | 6 [2.5–8] | 0.645 | 4 [3–5] | 4 [2–5.75] | 0.959 |

Notes.

The numbers in the table are presented in the following: mean ± standard deviation and median [25th percentile–75th percentile].

BMI, body mass index; SMI, skeletal muscle mass index.

**Table 3   Association between certification level deterioration and changes in usual walking speed using binomial logistic regression analysis.**

|  | β | Odds ratio | 95% CI | *P*-value |
|---|---|---|---|---|
| Model I |  |  |  |  |
| Change in usual walking speed | −4.558 | 0.010 | 0.000–0.422 | 0.016 |
| Model II |  |  |  |  |
| Change in usual walking speed | −3.964 | 0.019 | 0.000–0.873 | 0.042 |

Notes.
Dependent variable: Maintenance/Improvement group = 0; Deterioration group = 1.
Change in usual walking speed: usual walking speed at follow-up minus usual walking speed at baseline.
Model I: Non-adjusted.
Model II: Adjusted for sex, LTCI certification level, and intractable neurological disease.
CI, confidence interval; LTCI, long-term care insurance.

**Table 4   Association between certification level deterioration and changes in body fat percentage using binomial logistic regression analysis.**

|  | β | Odds ratio | 95% CI | *P*-value |
|---|---|---|---|---|
| Model I |  |  |  |  |
| Change in body fat percentage | −0.202 | 0.817 | 0.687–0.970 | 0.021 |
| Model II |  |  |  |  |
| Change in body fat percentage | −0.246 | 0.782 | 0.647–0.945 | 0.011 |

Notes.
Dependent variable: Maintenance/Improvement group=0; Deterioration group = 1.
Change in body fat percentage: body fat percentage at follow-up minus the body fat percentage at baseline.
Model I: Non-adjusted.
Model II: Adjusted for sex, LTCI certification level, and intractable neurological disease.
CI, confidence interval; LTCI, long-term care insurance.

changes in body fat percentage were significantly associated with the deterioration of LTCI certification levels in the non-adjusted ($\beta = -0.202$, $P = 0.021$) and adjusted models ($\beta = -0.246$, $P = 0.011$). The Hosmer–Lemeshow test result was $P \geq 0.05$ for all these models. For usual walking speed, the post-hoc statistical power was 0.89 in the non-adjusted model and 0.95 in the adjusted model. For body fat percentage, the post-hoc statistical power was 0.86 and 0.99 in the non-adjusted and adjusted models, respectively, which indicated good power. Tables S2 and S3 present the results of the binomial logistic regression analysis using only the deterioration and maintenance groups, excluding the improvement group. The ROC curves are shown in Fig. 2. The cut-off values to discriminate deterioration in LTCI certification levels were −0.14 m/s for change in the usual walking speed (sensitivity 85.7%, specificity 42.9%, AUC = 0.64, $P = 0.047$), and −1.0% for change in body fat percentage (sensitivity 72.7%, specificity 61.9%, AUC = 0.66, $P = 0.029$).

## DISCUSSION

In this study, handgrip strength, usual walking speed, and skeletal muscle mass, which are essential for assessing sarcopenia, showed that a decrease in usual walking speed over time reflected worsening certification levels, with a cutoff value of 0.14 m/s decrease. Previous studies have searched for factors in the occurrence of new LTCI certification by older adults who are not eligible for LTCI or in the deterioration of LTCI levels by institutionalized

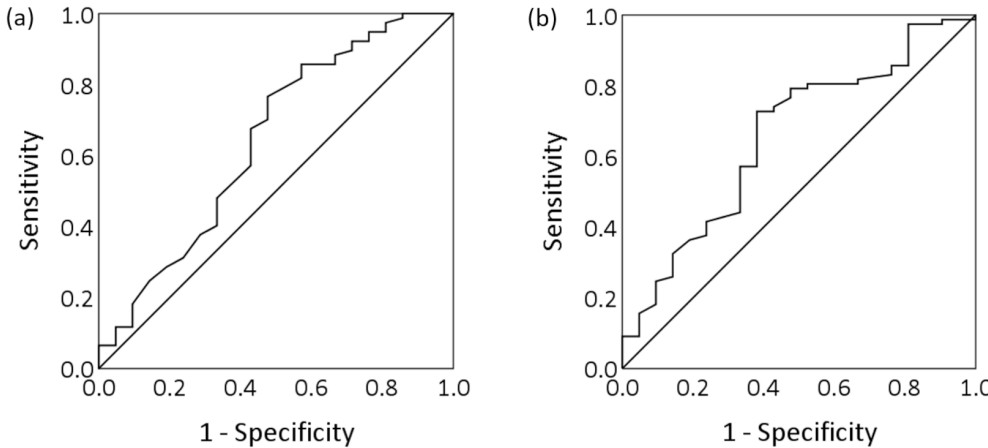

| | Cut-off | Sensitivity | Specificity | AUC | 95%CI | *P* value |
|---|---|---|---|---|---|---|
| (a) Usual walking speed | -0.14 m/sec | 85.7% | 42.9% | 0.64 | 0.50-0.79 | 0.047 |
| (b) Body fat percentage | -1.0% | 72.7% | 61.9% | 0.66 | 0.52-0.79 | 0.029 |

**Figure 2** **Receiver operating characteristic curve analysis using changes in the usual walking speed and body fat percentage to discriminate deterioration in long-term care insurance certification levels.** The amount of change refers to the change between baseline and follow-up over 6 months. AUC, area under the curve; CI, confidence interval; LTCI, long-term care insurance.

residents (*Shimada et al., 2023*; *Ashida, Kondo & Kondo, 2016*; *Shimizu, Ide & Kondo, 2023*; *Matsumoto & Takatori, 2021*; *Jin et al., 2018*). Our study is unique in that it focuses on older adults living "at home" while receiving LTCI services–that is, those whose functional ability is between that of healthy individuals who are not eligible for LTCI and those who live in institutions, and uses changes in LTCI certification levels as the main outcome. The three research questions described in the introduction section are discussed below.

Regarding the first research question, among the typical assessments of sarcopenia such as handgrip strength, walking speed, and skeletal muscle mass, usual walking speed was the assessment item that reflected deterioration in LTCI certification levels. Gait disturbance is an important factor in worsening prognosis (*Landi et al., 2010*), and a positive correlation exists between usual walking speed and life expectancy (*Studenski et al., 2011*). Usual walking speed predicts future LTCI incidents better than complex walking tasks (*Shimada et al., 2023*). These studies suggest a significant association between severe illness/need for LTCI and walking speed, supporting the results of this study. Moreover, the usual walking speed has the advantage of being simple and accurate for both the measurer and the examinee. The intraclass correlation coefficients of usual walking speed were 0.903 for a wide age range of 20–79 years (*Bohannon, 1997*), 0.97 for daycare users (*Chan & Pin, 2020*), 0.94 for residential care users (*Chan & Pin, 2020*), and 0.91 even for older patients with dementia (*Chan & Pin, 2019*), whose reliability was expected to be low. The reliability was high for individuals from various backgrounds. Furthermore, usual walking speed is unlikely to cause errors in the specific direction of fixed and proportional bias, even in older

adults requiring LTCI (*Sawaya et al., 2021*). Walking speed is an essential assessment tool for older adults because it is included in the assessment of frailty, which is multifactorial, and sarcopenia (*Chen et al., 2020*; *Satake & Arai, 2020*).

Regarding the second research question, the baseline and follow-up changes in usual walking speed, but not the baseline data, responded to the deterioration in LTCI certification levels. All participants had various disabilities requiring care and support, and had received LTCI certification. In other words, all participants already had a certain degree of functional decline, which may explain why the baseline muscle strength and physical function data were not characterized. In addition, the multivariate analysis showed that a decrease in usual walking speed was significantly associated with the deterioration of LTCI certification levels, even after adjustment. Thus, an intensive rehabilitation intervention targeting older adults requiring LTCI who show a decline in walking speed over time rather than a functional decline at baseline may contribute to preventing deterioration in the LTCI certification levels. Similarly, several studies focusing on changes over time reported that a decrease in usual walking speed is associated with an increased risk of mortality, supporting our findings (*Perera et al., 2005*; *White et al., 2013*).

Regarding the third research question, the change in usual walking speed that discriminated deterioration in LTCI certification levels was a decrease of 0.14 m/s. The minimal detectable change (MDC), which indicates the clinically relevant changes in the usual walking speed in older Japanese adults requiring LTCI, has been reported to be 0.18 m/s (*Sawaya et al., 2021*) and 0.19 m/s (*Takada & Tanaka, 2021*). A 1-year decrease of 0.10 m/s in the usual walking speed predicts death within 5 years (*Perera et al., 2005*); additionally, cutt-off of clinically significant interventions in the walking speed in patients with sarcopenia were observed to increase 0.10 m/s (*Morley et al., 2011*). The "clinically relevant changes" in usual walking speed in the previous and present studies were similar, and thus valid. Applying the cut-off value of 0.14 m/s to the walking distance, the decrease in usual walking speed during this period of deterioration in LTCI certification levels would be 0.56 s for 4 m, 0.7 s for 5 m, and 1.4 s for 10 m. A decrease in usual walking speed can be indicative of potential worsening of disability.Physiotherapists and other rehabilitation professionals should be attentive to these changes in clinical settings.

Finally, a decrease in body fat percentage was also identified as a factor contributing to deterioration in LTCI certification levels. This indicates that a decrease in body fat percentage is more sensitive to deterioration in LTCI certification levels than body weight or SMI. In eight studies that analyzed factors contributing to changes in certification levels among LTCI recipients, body fat percentage was not included as a predictor in all the studies (*Kumai & Ikeda, 2021*; *Ohnuma et al., 2016*; *Kamiya et al., 2017*; *Kawamura et al., 2021*; *Jin et al., 2018*; *Hayasaka et al., 2018*; *Kawamura, Kato & Kondo, 2018*; *Miyagishima et al., 2015*). To the best of our knowledge, this is the first study to report the association between changes in LTCI certification levels and body fat percentage. Body fat percentage is a nutritional index that reflects visceral fat mass in older adults (*Kupusinac et al., 2017*). A high body fat percentage may protect against sarcopenia (*Jackson et al., 2012*). The body fat percentage peaks at 80 years of age and begins to decline later than the muscle strength and skeletal muscle mass (*Jackson et al., 2012*; *Yoo, Won & Soh, 2022*; *Makizako et al., 2017*).

Therefore, it may be a sensitive indicator of the deterioration in LTCI certification levels among older adults who require LTCI.

This study has several limitations. First, we found no factors associated with improvements in the LTCI certification levels. The follow-up period in this study was relatively short. Second, the rehabilitation program in daycare was difficult to adjust because it was tailored to an individual's physical and mental functional status. Third, this study included only those whose body composition could be measured in the standing position and did not include those who had difficulty maintaining the standing position. Despite these limitations, this study is unique because it follows a transition in levels after LTCI certification, which is difficult to track. The results may help guide multidisciplinary interventions for older adults who continue to live in their home communities while undergoing LTCI.

## CONCLUSIONS

In conclusion, deterioration in LTCI certification levels was sensitively predicted not by baseline data but by decreases in usual walking speed and body fat percentage over time. The cut-off values were a 0.14 m/s decrease in the usual walking speed and a 1% decrease in the body fat percentage. Multidisciplinary rehabilitation interventions for older adults requiring LTCI who show a decrease in usual walking speed and body fat percentage may contribute to the prevention of worsening LTCI certification levels.

## ACKNOWLEDGEMENTS

We would like to thank all participants and staff involved in this study.

### Funding
This work was supported by the Japan Society for the Promotion of Science Grants-in-Aid (22K17539 and 23K06873). The funders had no role in study design, data collection and analysis, decision to publish, or preparation of the manuscript.

### Grant Disclosures
The following grant information was disclosed by the authors:
Japan Society for the Promotion of Science Grants-in-Aid: 22K17539 and 23K06873.

### Competing Interests
The authors declare there are no competing interests.

### Author Contributions
- Yohei Sawaya conceived and designed the experiments, performed the experiments, analyzed the data, prepared figures and/or tables, authored or reviewed drafts of the article, and approved the final draft.

- Tamaki Hirose conceived and designed the experiments, performed the experiments, authored or reviewed drafts of the article, and approved the final draft.
- Takahiro Shiba performed the experiments, authored or reviewed drafts of the article, and approved the final draft.
- Ryo Sato performed the experiments, authored or reviewed drafts of the article, and approved the final draft.
- Lu Yin performed the experiments, authored or reviewed drafts of the article, and approved the final draft.
- Akira Kubo performed the experiments, authored or reviewed drafts of the article, and approved the final draft.
- Tomohiko Urano conceived and designed the experiments, authored or reviewed drafts of the article, and approved the final draft.

## Human Ethics

The following information was supplied relating to ethical approvals (i.e., approving body and any reference numbers):

The study protocol was approved by International University Health and Welfare Ethics Review Board (approval numbers: 21-Io-22-2, 17-Io-189-7).

## Data Availability

The raw measurements are available in the Supplementary File.

## Supplemental Information

Supplemental information for this article can be found online at http://dx.doi.org/10.7717/peerj.17529#supplemental-information.

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
