# Peer review of "Decrease in the usual walking speed and body fat percentage associated with a deterioration in long-term care insurance certification levels"

_PeerJ, doi:10.7717/peerj.17529_

## Round 0.1 · original submission · Major Revisions

The study entitled “Decrease in the usual walking speed and body fat percentage associated with a deterioration in long-term care insurance certification levels” demonstrated excellent findings using an appropriate methodological approach. However, some important points must be clarified in the manuscript. Your article has great potential for publication on PeerJ, but the reviewers have requested substantial changes to be made in the text.

Reviewer 1 ·

Basic reporting

This study was conducted to identify sarcopenia-related assessments sensitive to transitions in Long-Term Care Insurance (LTCI) certification levels and determine cut-off values to predict them.The results of this study were interesting in that the deterioration group showed significant changes in the usual walking speed and body fat percentage. This suggests that decreases in usual walking speed and body fat percentage may predict worse certification levels in older adults requiring LTCI, which is considered a strength of this study. However, this study does not provide explanations and criteria for LTCI certification levels or the treatments received according to each level. Without such information, it may be difficult to clearly explain the results of this study. Additionally, it seems that further elaboration on the association of sarcopenia with strengthening LTCI grading criteria is necessary. Therefore, we have decided to classify this study as “ Major revision”." My comments are as follows. Thank you.

Experimental design

no comments

Validity of the findings

no comments

Additional comments

Major Comments
- In the introduction section, a more detailed explanation of LTCI is needed. For example, classification criteria, Benefits or characteristics associated with each certification level could include
- In the 82th line, about LTCI research, it was mentioned that there is a gap in terms of evaluating sarcopenia, but it does not align with the context. It seems that the explanation of sarcopenia should be provided first for such clarity to emergeIn the introduction part, there is insufficient explanation of sarcopenia, and it is necessary to explain why it is relevant to LTCI.
- In the 267th line, there should be a more detailed discussion on the relationship between body fat percentage and LTCI certification levels.
- An explanation is needed for why care level 5 was classified as 7.
- More detailed information about LTCI certification levels for each group needs to be provided. For example, the change in LTCI certification levels and the raw data for each group
- A detailed description of sarcopenia diagnosis should be provided in the Methods section

Annotated reviews are not available for download in order to protect the identity of reviewers who chose to remain anonymous.

·

Basic reporting

Initial comments: I would like to thank the authors and editor for the opportunity to review this important work for the scientific community and for this population group that still lacks research, which are the elderly who use long-term care (Long-Term Care Insurance).

Experimental design

Materials & Methods


Comment 04: This information: (mean age: 78.5 ± 8.1 years, 51 males and 47 females) is more appropriate in the results (line 111).
Comment 05: Including people who can stand when measuring body composition tests as an inclusion criterion is very vague. It makes me have countless questions, such as: Were people of both sexes included? From how old? Residents of urban or rural areas? Did elderly people with chronic illnesses participate in the research? I suggest reviewing these criteria.
Comment 06: In my opinion, the point “Long-term care insurance system” should come before “Changes in the long-term care insurance certification level”.
Comment 07: Regarding the “Long-term care insurance system”, it would be interesting for the reader if the authors explained that for this study, this variable was divided into 3 categories, but depending on the analysis, into only 2.

Validity of the findings

Results
Comment 08: I had doubts regarding data analysis. In tables 01 and 02, the authors initially carried out the analyzes considering 3 groups, however, during the analysis (specifically for tables 03 and 04), the variable was dichotomized. I would like to know why this recategorization and whether it would not be feasible to start the analysis with just two categories.
Comment 09: Indicate significant results in bold in the table.
Discussion
Comment 10: At the beginning of the discussion, I advise adding a brief summary of what was found in your results.
Comment 11: I didn't understand this sentence: “Regarding the first research question, the assessment of sarcopenia that responded most sensitively to deteriorating LTCI certification levels was usual walking speed.”(Line 225, page 10). What does it mean exactly?
Comment 12: Can the number of participants analyzed be considered a limitation?
Comment 13: I do not consider the AUC ROC results as a limitation of the work. Perhaps it is better to look for other points that the authors consider limitations that may have influenced the results.

Additional comments

My comments about the paper:

Abstract:
The information contained in the “Background” refers to the general objective of the study and does not provide a brief “introduction” of the topic. I suggest either adding information from the introduction or replacing Background with Objective.
Make it clear that this is a longitudinal study in terms of methods.
Introduction
Comment 01: Insert references to the information contained in lines 68-72 (page 2).
Comment 02: I note that the title of the work refers to gait speed and % body fat (which are the variables that showed significance in the analysis), however, the authors bring up sarcopenia as a gap in the research. I would like to understand exactly the introduction of sarcopenia in the study.
Comment 03: Still related to the comment above, I see that there is a lack of justification that explains why simpler sarcopenia screening tools should be used to change the level in LTCI. This need is not clear (to me) throughout the text, nor is the relationship between sarcopenia and older people who are included in this LTCI system evident.

---

## Round 0.2 · accepted · Accept

Congratulations! The authors have responded appropriately and effectively to the comments.

Reviewer 1 ·

Basic reporting

The author responded appropriately and effectively to my comments, which I believe has improved the overall quality of the research paper. Therefore, my decision is to accept it. Thank you.

Experimental design

single-center prospective cohort study

Validity of the findings

Decreases in usual walking speed and body
fat percentage may predict worse certiûcation levels in older adults requiring LTCI.

·

Basic reporting

Congratulations to the authors. I consider that all the considerations I made in the first review were addressed. And a significant improvement in the work can be observed. In this version, I was able to understand some points that had been a subject of questioning the first time.

Experimental design

no comment

Validity of the findings

no comment

Additional comments

no comment